

# The relationship between virtual simulation, critical thinking, and self-directed learning abilities of nursing students in Riyadh, Saudi Arabia

Hanan F. Alharbi[1], Amjad Alsubaie[2], Rahaf Gharawi[2], Rawan Ba Mazroo[2], Shaikhah Alajaleen[2], Munerah Alsultan[2], Munira Alsaleem[2], Nora Alsubihi[2], Norah Alsahli[2], Nashwa Alqahtani[2] and Raghad Rayzah[2]

[1] Maternity and Pediatric Nursing Department, College of Nursing, Princess Nourah bint Abdulrahman University, Riyadh, Saudi Arabia
[2] College of Nursing, Princess Nourah bint Abdulrahman University, Riyadh, Saudi Arabia

## ABSTRACT

**Objective**. The use of virtual simulation in nursing education is an effective approach for improving nursing critical thinking and self-learning abilities, but the previous studies were limited to providing the required evidence that supports the association. This study aimed to assess the relationship between virtual simulation and critical thinking disposition and self-directed learning abilities among nursing students.

**Methods**. This is a descriptive correlational, non-experimental study. It was conducted among 201 third- and fourth-year nursing students at the Academic Institution, Saudi Arabia. A non-probability convenience sampling technique was used to select the participants; then, an online, adapted questionnaire was sent to the participants, the data from which was analyzed by SPSS.

**Results**. The study findings showed that virtual simulation benefited nursing students. Most participants (56%) agreed that it helped them to think critically, and approximately 27% strongly agreed. It also enhanced their self-directed learning abilities, and the majority of the students agreed that they often review the way nursing practice is conducted. Furthermore, the results showed a significant, positive relationship between virtual simulation and the critical thinking disposition of nursing students ($p$-value = 0.03; correlation coefficient = 0.65), a strong positive relationship with self-directed learning abilities of nursing students ($p$-value = 0.004; correlation coefficient = 0.78), and a strong positive relationship between critical thinking disposition with self-directed learning abilities of nursing students ($p$-value = 0.01; correlation coefficient = 0.72).

**Conclusion**. There are significant relationships between virtual simulation and the critical thinking disposition and self-directed learning abilities of nursing students. Furthermore, virtual simulation made the students practice critical thinking and self-learning, so, they simulate events and try to seek out and solve the problems.

Corresponding author
Hanan F. Alharbi,
hfalharbi@pnu.edu.sa

## INTRODUCTION

Recent developments in healthcare systems necessitate that nurses are well-prepared to handle complex clinical situations, particularly in the post-COVID-19 era. Recent developments in healthcare systems require that nurses are well prepared to decomplex situations, particularly in the post-COVID-19 era (*Anders, 2021*). So, it is necessary to prepare nursing students to be competent in advanced technology skills, critical thinking disposition, and self-directed learning (SDL) abilities (*Arizo-Luque et al., 2022*; *Kim & Shin, 2021*). It was found that self-directed leaning in more effective in knowledge acquisition among nursing student in Saudi Arabia (*Ahmed, Alostaz & AL-Lateef Sammouri, 2016*)

Previous studies, such as a study conducted by *Shin & Kim (2014)*, have illustrated that using simulations in nursing education effectively improve critical thinking skills in various fields, including pediatrics, and the multiple exposures for simulation in the nursing education coursework results in the gain of critical thinking skills and its subcategories (*Shin et al., 2015*). Furthermore, virtual simulation allows nursing students to practice their skills in an environment that is close to a clinical setting without affecting patient safety. Literature indicates that virtual simulation is an efficient method that poses no risks to patient safety (*Ezzeddine, 2017*).

In nursing education, directed self-learning (DSL) allows learners to find significance and reason in their learning and to take responsibility for the preparation, execution, and assessment of their learning (*Lee, Kim & Chae, 2020*). It generates a significant improvement in the level of knowledge acquisition and overall performance (*Ahmed, Alostaz & AL-Lateef Sammouri, 2016*; *Abdullah et al., 2018*). A study conducted by *Bodur et al. (2024)* showed that nursing students expressed positive view regarding virtual learning and improved their self-directed learning skills.

Several methods have been used by nursing educators to grow critical thinking among students. Virtual simulation is one of these utilized methods (*Blakeslee, 2020*) and provides an exceptional learning opportunity for nursing students with the significant achievement of the learning outcomes (*Foronda et al., 2020*). It was reported by *Song & Kim (2023)* that nursing students satisfaction regarding virtual simulation in nursing students.

Virtual simulation in nursing education has provided an enhancement in nursing students' learning abilities, as illustrated in an integrative review of the educational characteristics related to virtual simulation in the field of nursing education by *Shin et al. (2019)* and another study by *Foronda et al. (2020)* reported that virtual simulation is an excellent educational tool for encouraging the learning process to reach the desired outcomes. Still, the previous studies were limited to providing solid evidence to support the association between virtual simulation and students' critical thinking or self-learning abilities (*Cant & Cooper, 2010*; *Chang et al., 2022*).

Through the use of virtual simulations, nursing students can hone their critical thinking skills and SDL abilities. These simulated environments provide those students with various challenges and scenarios to explore and analyze (*Warren et al., 2016*; *Jin & Ji, 2021*). Students are actively encouraged to think independently and creatively as they work through the simulations. This learning environment helps nurture their capacity to think

critically and develop the skills necessary to face the real world (*Chan, 2019*; *Yildirim, Özkahraman & Karabudak, 2011*). Additionally, it allows students to create a sense of self-discipline and SDL, as they can work through the simulations independently. With virtual simulation, nursing students can develop the skills they need to become successful in the future (*Thiagraj, Karim & Veloo, 2021*; *Bruce, Levett-Jones & Courtney-Pratt, 2019*). Furthermore, a study conducted by *Yeo & Jang (2023)* showed that the virtual environment has developed confidence, achievement, and satisfaction among the learners in addition to correcting their errors from failure.

There are different studies conducted across the globe on simulation effectiveness in the medical field. However, there are not many studies done on virtual simulation effectiveness in the nursing field, including the Middle Eastern countries; therefore, this study aimed to assess the relationship between virtual simulation with critical thinking and the SDL of nursing students at the Academic Institution, Riyadh, Saudi Arabia.

# MATERIALS AND METHODS

## Study design

This is a descriptive correlational, non-experimental study. It is one of the descriptive study designs used to assess the relationships between two or more factors (*Lau, 2017*)

## Study setting

This study was done at the College of Nursing, Academic Institution, Riyadh, Saudi Arabia.

## Study population

The population included in the current study is the third- and fourth-year nursing students at the Academic Institution, Riyadh, Saudi Arabia. The total number of students registered in third and fourth year of nursing program for the academic year 2022/2023 were invited to participate in the study. Of those invited, (250 students), a number of 201 responded to the questionnaire which resulted in a participation rate of 80.4%.

The inclusion criteria were third and fourth-year nursing students from the Academic Institution who experienced simulation at least once. The exclusion criteria were the first and second-year nursing students at the Academic Institution who did not experience the simulation. The simulation used in this study was virtual high-fidelity through the university simulation center. It included an interactive platform and virtual scenarios for nursing interventions, it allows students to understand nursing procedures and develop their learning, and critical thinking skills.

## Sampling and sample size

The sampling technique in the present research was non-probability convenience sampling, and the sample size in the study was 201, which was calculated based on the Epi Info program calculation. The level of confidence is 99%, and the sampling error is 1%.

## Data collection technique

The data was collected on critical thinking disposition and SDL abilities from nursing students who experienced virtual simulation at least once. The data were gathered using
a google forms questionnaire distributed in various ways. Afterward, the aim and content of the study were clarified to the participants, and informed consent was collected before data collection.

## Instruments of data collection

This study has adopted three scales for measuring virtual simulation, critical thinking disposition, and SDL abilities; the first tool was for virtual simulation, which was adopted from a tool used by Ryan-Wenger, Elfrink Cordi, Leighton, Doyle, and Ravert (2012). It comprises 13 sub-items with a score of 0, 1, or 2 (*Cordi et al., 2012*). The second tool was for critical thinking disposition, which has 27 sub-items validated by *Yoon (2004)*. The third tool was adopted from *Fisher, King & Tague (2001)* to measure SDL abilities and is composed of 40 items.

## Tools validity

The reliability and validity test has been applied to the simulation effectiveness tool to determine the extent to which items in the tool were related to each other by Cronbach's co-efficiency Alpha ($a = 0.954$). Therefore, it can be concluded that the tool has a high level of reliability. Test of the instrument's validity was conducted using Pearson Product Moment Correlations using SPSS. The significant value obtained by the Sig (2-tailed) <0.05 and the internal consistency ($r = 0.890$, *p*-value < 0.001) indicated that the items of the tool were valid.

The reliability of the critical thinking disposition tool was done to determine the extent to which items in the tool were related to each other by Cronbach's alpha ($a = 0.946$). Therefore, the tool has a high level of reliability. Test the validity of the instrument was conducted using Pearson product-moment correlations using SPSS. From the significant value obtained by the Sig (2-tailed) <0.05 and the internal consistency ($r = 0.962$, *p*-value < 0.001), the items of the tool were considered valid.

The reliability of the SDL ability scale tool was done to determine the extent to which items in the tool were related to each other by Cronbach's alpha ($a = 0.0.971$). The tool results indicate a high level of reliability. The instrument's validity was tested using Pearson's product-moment correlations using SPSS. Based on the significant value obtained by the Sig (2-tailed) <0.05 and the internal consistency ($r = 0.980$ *p*-value < 0.001), so it can be concluded that items of the tool were valid.

## Statistical analysis

The data were analyzed by the Statistical Package for Social Science (SPSS)-version 22 with Microsoft Excel program. The findings were presented as mean and standard deviation (X ± SD) or as frequency and percentage (no. and %).

## Ethical considerations

Ethical approval from the Institution of Review Board (IRB) approval was obtained from the Academic Institution (No. H-01-R-059, 21-0043), Riyadh, Saudi Arabia. The participants' written consent to take part in the research was obtained prior to their participation. The students were informed of their right to withdraw from the study at any time.

**Table 1  Demographic characteristics of the studied students ($n = 201$).**

| Items | | No. | % |
|---|---|---|---|
| Academic year | Third year | 102 | 50.7% |
| | Fourth year | 99 | 49.3% |
| Gender | Female | 201 | 100% |
| Age | 20 years | 75 | 37.3% |
| | 21 years | 126 | 62.7% |
| Nationality | Saudi | 201 | 100% |

**Table 2  Distribution of simulation effectiveness of the studied students ($n = 201$).**

| Items | Strongly agree | | Agree | | Neither | | Disagree | | Strongly disagree | |
|---|---|---|---|---|---|---|---|---|---|---|
| | No. | % | No. | % | No. | % | No. | % | No. | % |
| The instructor's questions helped me to think critically | 55 | 27.4% | 112 | 55.7% | 15 | 7.5% | 8 | 4.0% | 11 | 5.5% |
| I was challenged in my thinking and decision-making skills | 63 | 31.3% | 106 | 52.7% | 12 | 6.0% | 12 | 6.0% | 8 | 4.0% |
| I developed a better understanding of the pathophysiology of the conditions in the simulated clinical experience | 50 | 24.9% | 89 | 44.3% | 33 | 16.4% | 16 | 8.0% | 13 | 6.5% |
| I am able to better predict what changes may occur with my real patients | 52 | 25.9% | 115 | 57.2% | 15 | 7.5% | 8 | 4.0% | 11 | 5.5% |
| I developed a better understanding of the medications that were in the simulated clinical experience | 52 | 25.9% | 108 | 53.7% | 21 | 10.4% | 9 | 4.5% | 11 | 5.5% |

# RESULTS

## Results summary

Table 1 shows the demographic characteristics of the studied nursing students; their academic year was approximately similar in the two years. Nearly 51% of the participants were third-year students, and 49% of the participants were fourth-year students. They were all female and Saudi; their age was either 20 or 21 years.

Table 2 represents the distribution of simulation effectiveness of the studied students. Participants were asked if the simulation experience instructor's questions helped them to think critically. Approximately 56% of participants agreed, and 27% strongly agreed with the statement. Nearly 53% of the participants agreed that the simulation experience challenged their thinking and decision-making skills and helped them develop a better understanding of the issues related to medication "effects, side effects, *etc*". More than half the participants agreed that they developed an understanding of the pathophysiology of the simulated condition, unlike the 8% who disagreed. 57% of the participants agreed, and 26% strongly agreed that the simulation helped them to predict changes that might happen to real patients.

Table 3 presents the critical thinking disposition of the studied participants. A total of 59% agreed that they look for a piece of information to solve a problem, that they tend to solve problems by a collection of data and a systematic organization, and that they have a reputation for being rational. A total of 56% agreed that they willingly solve problems and explain the reasons if they do not agree with others, and 54% agreed that they judge

**Table 3** Distribution of critical thinking disposition of the studied students (n = 201).

| Items | Strongly agree | | Agree | | Neither | | Disagree | | Strongly disagree | |
|---|---|---|---|---|---|---|---|---|---|---|
| | No. | % | No. | % | No. | % | No. | % | No. | % |
| I continually look for pieces of information related to solving a problem | 47 | 23.4% | 119 | 59.2% | 20 | 10.0% | 6 | 3.0% | 9 | 4.5% |
| I willingly solve a complicated problem | 34 | 16.9% | 113 | 56.2% | 28 | 13.9% | 16 | 8.0% | 10 | 5.0% |
| I'm trying to understand how the unknown things work | 58 | 28.9% | 111 | 55.2% | 12 | 6.0% | 9 | 4.5% | 11 | 5.5% |
| When I confront a problem, I try hard to find an answer until solving it | 68 | 33.8% | 105 | 52.2% | 13 | 6.5% | 8 | 4.0% | 7 | 3.5% |
| I explain reasons if I don't agree with others | 60 | 29.9% | 114 | 56.7 | 17 | 8.5% | 5 | 2.5% | 5 | 2.5% |
| When I am questioned, I think twice before I give my answer | 61 | 30.3% | 97 | 48.3% | 20 | 10.0% | 14 | 7.0% | 9 | 4.5% |
| I don't rush to judgment | 45 | 22.4% | 95 | 47.3% | 34 | 16.9% | 14 | 7.0% | 13 | 6.5% |
| I continually evaluate whether my thought is right or not | 48 | 23.9% | 110 | 54.7% | 20 | 10.0% | 16 | 8.0% | 7 | 3.5% |
| When I see the world, I see it with a questioning mind | 63 | 31.3% | 102 | 50.7% | 18 | 9.0% | 10 | 5.0% | 8 | 4.0% |

objectively. Over 80% agreed that they are trying to know how unknown things work, with half of the participants agreeing that they see the world with a questioning mind. 52% agreed that when they confront a problem, they try hard to find an answer until a solution is found. More than half agreed that they continually evaluate whether their thought is right or not. The majority agreed that when they are questioned, they think twice before answering, while 47% agreed that they do not rush to judgment.

Table 4 reflects the distribution of the SDL ability of the studied students. The majority of the participants in the study agree with 57% that they often review the way nursing practice is conducted. 57% of participants agreed that they evaluated their performance. Nearly 53% agreed that they are open to new learning opportunities, while nearly 52% agreed that they prefer to direct their learning. Almost 50% agreed that they enjoy learning new information and need minimal help to find information. About 54% agreed that they can find out information for themselves and they critically evaluate new ideas. About half the participants agreed that they would ask for help in learning when necessary and they would learn from their mistakes. Almost all the participants agreed that they need to know why, "the rationale".

Table 5 demonstrates the level of simulation effectiveness. Most of the participants had high to moderate levels of simulation effectiveness. Ninety-three participants (46%) had a high level of simulation effectiveness, and ninety-four of the participants, nearly 47%, had a moderate level of simulation effectiveness. Only 7% of participants reported a low level of simulation effectiveness. It also showed the level of critical thinking disposition of the studied students. The majority of the participants had a moderate level of critical thinking disposition. There were 32% who had a high level of critical thinking disposition, and 64% had a moderate level of critical thinking disposition. A minority of participants revealed a low level of critical thinking disposition. The level of SDL ability scale of the studied students. Nearly half (n = 99) had a high level of SDL abilities, and 97 participants

**Table 4 Distribution of SDL abilities of the studied students ($n = 201$).**

| Items | Strongly agree | | Agree | | Neither | | Disagree | | Strongly disagree | |
|---|---|---|---|---|---|---|---|---|---|---|
| | No. | % | No. | % | No. | % | No. | % | No. | % |
| I am open to new learning opportunities | 63 | 31.3% | 106 | 52.7% | 21 | 10.4% | 2 | 1.0% | 9 | 4.5% |
| I enjoy learning new information | 76 | 37.8% | 100 | 49.8% | 12 | 6.0% | 5 | 2.5% | 8 | 4.0% |
| I evaluate my own performance | 70 | 34.8% | 114 | 56.7% | 13 | 6.5% | 0 | 0.0% | 4 | 2.0% |
| I can find out information for myself | 65 | 32.3% | 108 | 53.7% | 15 | 7.5% | 3 | 1.5% | 10 | 5.0% |
| I need minimal help to find information | 52 | 25.9% | 100 | 49.8% | 30 | 14.9% | 12 | 6.0% | 7 | 3.5% |
| I prefer to plan my own learning | 63 | 31.3% | 105 | 52.2% | 23 | 11.4% | 3 | 1.5% | 7 | 3.5% |
| I prefer to direct my own learning | 62 | 30.8% | 104 | 51.7% | 25 | 12.4% | 4 | 2.0% | 6 | 3.0% |
| I often review the way nursing practices are conducted | 52 | 25.9% | 115 | 57.2% | 20 | 10.0% | 8 | 4.0% | 6 | 3.0% |
| I need to know why | 70 | 34.8% | 111 | 55.2% | 11 | 5.5% | 3 | 1.5% | 6 | 3.0% |
| I critically evaluate new ideas | 68 | 33.8% | 108 | 53.7% | 12 | 6.0% | 9 | 4.5% | 4 | 2.0% |
| I will ask for help in my learning when necessary | 74 | 36.8% | 97 | 48.3% | 15 | 7.5% | 6 | 3.0% | 9 | 4.5% |
| I learn from my mistakes | 78 | 38.8% | 102 | 50.7% | 15 | 7.5% | 2 | 1.0% | 4 | 2.0% |

**Table 5 Level of simulation effectiveness, CT disposition and SDL abilities of nursing students at PNU.**

| Variable | Items | No. | % |
|---|---|---|---|
| **Level of simulation effectiveness** | Low | 14 | 7% |
| | Moderate | 94 | 46.8% |
| | High | 93 | 46.3% |
| **Level of critical thinking disposition** | Low | 7 | 3.50% |
| | Moderate | 129 | 64.20% |
| | High | 65 | 32.30% |
| **Level of self-directed learning ability** | Low | 5 | 2.5% |
| | Moderate | 97 | 48.3% |
| | High | 99 | 49.3% |

(48%) had a moderate level of self-learning ability. The minority reported a low level of SDL abilities.

Table 6 shows the relationships between virtual simulation, critical thinking skills, and SDL abilities of nursing students at the Academic Institution. A positive, strong relationship between virtual simulation and the critical thinking disposition of nursing students was found with a significant correlation ($p$-value = 0.03; correlation coefficient = 0.65), a positive, strong relationship between virtual simulation and the SDL abilities of nursing students was observed with significant correlation ($p$-value = 0.004; correlation coefficient = 0.78), and a positive strong relationship between critical thinking disposition and SDL abilities of nursing students was observed with significant correlation ($p$-value of 0.01; correlation coefficient = 0.72).

**Table 6  The relationship between virtual simulation, critical thinking skills, and SDL abilities of nursing students at PNU.**

|  | Virtual simulation | Critical thinking disposition | SDL abilities |
|---|---|---|---|
|  | r(p) | r(p) | r(p) |
| **Virtual simulation** | 1 |  |  |
| **Critical thinking disposition** | 0.65 (0.03) | 1 |  |
| **SDL abilities** | 0.78 (0.004) | 0.72 (0.01) | 1 |

## DISCUSSION

Virtual simulation is becoming increasingly popular as a learning tool in nursing education. It can be used to teach and assess critical thinking disposition and SDL abilities in nursing students, as well as provide a safe environment to practice clinical decision-making. This study assessed the relationship between virtual simulation with critical thinking skills and SDL abilities of nursing students at the Academic Institution.

This study explored the relationship of virtual simulation with nursing students' critical thinking disposition and SDL abilities. The findings indicate a significant correlation between virtual simulation and both the critical thinking and SDL abilities of nursing students. A similar study conducted in Indonesia investigated the impact of virtual simulation on critical thinking and found that critical thinking was strengthened and increased with virtual simulation (*Ikhsan, Sugiyarto & Astuti, 2020*), and the Spanish simulation environment has resulted in improvement in critical thinking abilities of nursing students (*Arizo-Luque et al., 2022*). On the other hand, a Korean study about the extent of the simulation effect on the self-learning ability of nursing students conducted by *Cho & Hwang (2019)* concludes that there are little to no differences and the same level of ability of self-learning among nursing students was maintained after the simulation (*Cho & Hwang, 2019*) and on the impact of virtual simulation on critical thinking and SDL which showed no statistically significant difference were observed after application of virtual simulation, and there is no increase or decrease with the existence of virtual simulation (*Kang, Hong & Lee, 2020*).

The simulation helps nursing students to gain confidence in their abilities and to learn how to respond to various situations. It can also help them to develop problem-solving skills and critical thinking skills, as well as to become better communicators (*Goulding et al., 2020*). It can also help reduce stress and anxiety in nursing students, allowing them to practice their skills and knowledge in a safe environment before facing real-life situations. Simulation is also an effective way to teach students tasks of prioritization and teamwork (*O'Flaherty & Costabile, 2020*). In the current study, the top-ranked item of simulation benefit was that it helped students to predict changes that may happen to a real patient; they learned the health assessments and performed them in laboratories in nursing schools. However, virtual simulations offer the possibility to practice with real cases (*Sundler, Pettersson & Berglund, 2015*; *Abelsson & Bisholt, 2017*). The virtual nursing simulation includes an assessment that students may have found useful for their assessment knowledge and skills (*Padilha et al., 2019*). The third-ranked item on

simulation effectiveness was that participating in simulation helped students to develop a better understanding of the medication that was in the simulated clinical experience. The goals of planning and applying simulation education are to help students to better the content of the class and to become better in clinical practice (*Dubovi, Levy & Dagan, 2017*). In this sense, the application of virtual simulation to nursing science has been successful for the student learning experience (*Sitterding et al., 2019*).

The critical thinking skills and SDL abilities improved significantly in the current study, and scores on one sub-scale of SDL abilities, "gathering resources for learning", also significantly correlated with virtual simulation. Those findings implicate that virtual simulation resources for nursing interventions could be a proactively effective approach in nursing education. This is a very relevant finding because of the need to exert continuous effort to seek nursing resources and knowledge to improve care for patients by nursing professionals. Furthermore, a similar study reported that nursing professionalism implicates searching and finding relevant information. The competence related to professional nursing ability was enhanced by virtual simulation in this current study (*Ghadirian, Salsali & Cheraghi, 2014*).

This study has several limitations; it was done only on female participants and in one university. Thus, the results cannot be generalized. More extensive research with different educational institutions and larger samples, including females and males, is warranted. It was also limited to examine the correlational relationship between virtual simulation with critical thinking disposition and SDL abilities among nursing students using descriptive design, which needs experimental study approach. Further, replication of this study using other nursing student samples, like those found in various universities and public higher education institutions, would aid in generalizing findings. Another limitation was that this study used non-probability convenience sampling techniques to select the students from one place, which restricts its generalizability.

## CONCLUSION

In conclusion, the study showed significant relationships between virtual simulation with critical thinking disposition and SDL abilities among nursing students. Besides, virtual simulation affected nursing students and made them develop critical thinking and self-learning abilities so that they simulate events and try to seek out and solve the problems. Most of the students stated that it had benefited their learning and that they experienced an environment close to a clinical setting. Virtual simulation could be helpful to nursing educators to maximize learning among students due to its education characteristics, provision of opportunities to synthesize nursing knowledge, and effective learning environment.

### Funding

This work is supported by Princess Nourah bint Abdulrahman University Researchers Supporting Project number (PNURSP2024R441), Princess Nourah bint Abdulrahman University, Riyadh, Saudi Arabia. The funders had no role in study design, data collection and analysis, decision to publish, or preparation of the manuscript.

### Grant Disclosures

The following grant information was disclosed by the authors:
Princess Nourah bint Abdulrahman University: PNURSP2024R441.

### Competing Interests

The authors declare there are no competing interests.

### Author Contributions

- Hanan F. Alharbi conceived and designed the experiments, performed the experiments, analyzed the data, prepared figures and/or tables, authored or reviewed drafts of the article, and approved the final draft.
- Amjad Alsubaie conceived and designed the experiments, performed the experiments, authored or reviewed drafts of the article, and approved the final draft.
- Rahaf Gharawi conceived and designed the experiments, performed the experiments, authored or reviewed drafts of the article, and approved the final draft.
- Rawan Ba Mazroo conceived and designed the experiments, performed the experiments, authored or reviewed drafts of the article, and approved the final draft.
- Shaikhah Alajaleen conceived and designed the experiments, performed the experiments, authored or reviewed drafts of the article, and approved the final draft.
- Munerah Alsultan conceived and designed the experiments, performed the experiments, authored or reviewed drafts of the article, and approved the final draft.
- Munira Alsaleem conceived and designed the experiments, performed the experiments, authored or reviewed drafts of the article, and approved the final draft.
- Nora Alsubihi conceived and designed the experiments, performed the experiments, authored or reviewed drafts of the article, and approved the final draft.
- Norah Alsahli conceived and designed the experiments, performed the experiments, authored or reviewed drafts of the article, and approved the final draft.
- Nashwa Alqahtani conceived and designed the experiments, performed the experiments, authored or reviewed drafts of the article, and approved the final draft.
- Raghad Rayzah conceived and designed the experiments, performed the experiments, authored or reviewed drafts of the article, and approved the final draft.

### Ethics

The following information was supplied relating to ethical approvals (i.e., approving body and any reference numbers):

Ethical approval (No. H-01-R-059, 21-0043) from the Institution of Review Board (IRB) approval was obtained from PNU, Riyadh, Saudi Arabia.

## Data Availability

The raw data is available in the Supplemental File.

## Supplemental Information

Supplemental information for this article can be found online at http://dx.doi.org/10.7717/peerj.18150#supplemental-information.

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
