# Peer review of "The relationship between virtual simulation, critical thinking, and self-directed learning abilities of nursing students in Riyadh, Saudi Arabia"

_PeerJ, doi:10.7717/peerj.18150_

## Round 0.1 · original submission · Major Revisions

Please look over the reviewer feedback and make edits to try and improve your manuscript. Please provide a response document that clearly outlines what changes were made in response to reviewer feedback. In instances where changes were not made in response to reviewer feedback please provide a rebuttal to justify the decision.

I note that it appears that reviewer 1 has made use of generative AI to help produce their review which has resulted in some feedback that is fairly vague. I note that there are more specific comments embedded throughout their feedback that looks more useful. I recommend focussing on the comments that are more specific in nature, where it is more apparent what can be done to address the comment/s. I believe it to be unfair to authors to require a point-by-point response to that type of review, so instead just make use of what feedback is useful, and make clear in the resubmission what changes have been made, and no need to provide rebuttals for what was ignored from R1.

For R2 and R3 feedback, please provide standard point by point response to their feedabck with changes that were made, and changes that were not made, in response to feedback.

On my own read of the paper, something I personally noticed is that a limitation should be acknowledged that correlational research like what has been reported in this paper should be backed up by more experimental research in the future that can more definitely examine the effectiveness of virtual simulation experiences for fostering critical thinking. This is because an alternative interpretation of your findings could be that students with higher critical thinking disposition simply provide more favourable attitutdes towards simulation activities, rather than the simulation experience predicting their critical thinking ability.

·

Basic reporting

Feedback on Introduction
Clarity and Professional Language
Clarity: The introduction is mostly clear and provides a good overview of the context, the importance of the study, and the gap in existing research. However, there are areas where clarity can be improved.
Unambiguous Language: The language is generally unambiguous, but there are instances where sentences can be rephrased for better understanding.
Professional English: The introduction is written in professional English but can benefit from minor grammatical and stylistic improvements.
Original Primary Research and Scope
Original Research: The introduction establishes that the study aims to address a gap in the existing literature, indicating original primary research. This fits within the scope of a journal focused on nursing education and technology in healthcare.
Scope of the Journal: The focus on virtual simulation in nursing education is relevant and timely, especially given the ongoing advancements in healthcare technology and the need for competent nursing professionals.
Context and Background
Context: The introduction effectively sets the context by highlighting the importance of preparing nursing students with critical thinking and self-directed learning (SDL) abilities, especially in the post-COVID-19 era.
Literature Reference: The introduction references relevant studies and literature, providing a solid background for the current research.
Suggestions for Improvement
Refine Sentence Structure:
Example: "The recent development in healthcare systems requires nurses to be well prepared to deal appropriately with complicated clinical circumstances in the work settings, especially in the era of COVID-19 and forward."
Improved: "Recent developments in healthcare systems necessitate that nurses are well-prepared to handle complex clinical situations, particularly in the post-COVID-19 era."
Clarify Purpose and Scope:
Make the purpose of the study more explicit towards the end of the introduction.
Example: "the current study was designed to assess the relationship between virtual simulation with critical thinking and the SDL of nursing students at PNU, Riyadh, Saudi Arabia."
Improved: "Therefore, this study aims to assess the relationship between virtual simulation and both critical thinking disposition and self-directed learning abilities among nursing students at Princess Nourah bint Abdulrahman University (PNU) in Riyadh, Saudi Arabia."
Ensure Smooth Flow:
Ensure each paragraph logically follows the previous one, maintaining a clear narrative flow.
Example: "Several previous studies have approved that high-fidelity simulation in nursing education effectively enhanced nursing students' critical thinking in different nursing fields, such as pediatrics, as shown in one study by Shin and Kim 2014."
Improved: "Previous studies, such as Shin and Kim (2014), have demonstrated that high-fidelity simulations in nursing education effectively enhance critical thinking skills in various fields, including pediatrics."
Minor Grammatical Corrections:
Example: "Furthermore, virtual simulation helps nursing students to practice nursing skills in an environment that is close to a clinical setting without affecting patient safety."
Improved: "Furthermore, virtual simulation allows nursing students to practice their skills in an environment that closely mimics a clinical setting without compromising patient safety."
Enhanced Literature Integration:
Integrate references more smoothly into the text to avoid breaking the flow.
Example: "According to the literature, virtual simulation is an efficient method that carries no risks to patient safety."
Improved: "Literature indicates that virtual simulation is an efficient method that poses no risks to patient safety (Shin et al., 2019)."
• The significance and rational beyond conducting this study is missed.
• Problem statement and the gap that this study will fulfill is not strongly enough stated in the introduction.
Feedback on the Discussion
Clarity and Grammatical Correctness
The discussion generally follows a logical structure and conveys the findings of the study effectively. However, it could benefit from clearer sentence structures and more precise language. Below are detailed suggestions for improvement:
Clarify Sentence Structure:
Example: "Based on the findings, it appears that both critical thinking and SDL of the nursing students were significantly correlated with virtual simulation."
Improved: "The findings indicate a significant correlation between virtual simulation and both the critical thinking and self-directed learning (SDL) abilities of nursing students."
Ensure Consistent Tense Usage:
Example: "A similar study has been conducted in Indonesia to investigate the impact of virtual simulation on critical thinking, and it showed that critical thinking was strengthened and increased with the virtual simulation."
Improved: "A similar study conducted in Indonesia investigated the impact of virtual simulation on critical thinking and found that critical thinking was strengthened and increased with virtual simulation."
Remove Redundant Phrases:
Example: "The simulation helps nursing students to gain confidence in their abilities and to learn how to respond to various situations."
Improved: "Simulation helps nursing students gain confidence and learn how to respond to various situations."
Address Repetition and Flow:
Example: "Simulation is also an effective way to teach students how to prioritize tasks and how to work as part of a team."
Improved: "Simulation is an effective way to teach students task prioritization and teamwork."
Improve Clarity of Findings:

Example: "In the current study, the top-ranked item of simulation benefit was that it helped students to predict changes that may happen to a real patient; they learned the health assessments and performed them in laboratories in nursing schools."
Improved: "In the current study, the highest-ranked benefit of simulation was its ability to help students predict changes in real patients, enhancing their health assessment skills practiced in laboratory settings."
Drawbacks of the Used References
Outdated Sources:
Some references are relatively old, such as those from 2001 and 2004 (references 22 and 23). The field of nursing education and simulation technology evolves rapidly, and newer studies might offer more current insights and data.
Limited Geographical Diversity:
The references mainly focus on studies conducted in specific regions like Korea, Indonesia, Spain, and Saudi Arabia (references 2, 4, 25, and 26). This might limit the generalizability of the findings to a global context. Including studies from a wider range of countries could provide a more comprehensive view.
Single Study Authors:
Several references involve the same primary authors (e.g., Shin H. appears in references 4, 5, 12). Relying heavily on a few researchers may introduce bias or limit the diversity of perspectives in the literature review.
Inconsistencies in Reference Formats:
There are inconsistencies in how the references are cited (e.g., some use full names, others use initials; some include full titles, others do not). This can affect the professionalism and readability of the reference list. Many studies had no DOI number
Lack of Directly Relevant Studies:
While many references discuss simulation or critical thinking, not all directly address the specific combination of virtual simulation, critical thinking, and self-directed learning in nursing education. More targeted references could strengthen the literature review.
Non-peer-reviewed Sources:
Some references, such as those listed as dissertations (reference 22), may not have undergone rigorous peer review. Peer-reviewed journal articles typically provide more reliable and validated findings.
Inadequate Evidence for Certain Claims:
The references used to support specific claims about virtual simulation’s effectiveness (references 24, 25, 26) show mixed results. It might be beneficial to include more studies that specifically demonstrate a significant positive impact to reinforce the study's findings.

Suggestions for Improvement
Update Sources:
Incorporate more recent studies (post-2020) to reflect the latest advancements and trends in virtual simulation and nursing education.
Diversify Geographic Representation:
Include studies from a wider range of countries and regions to ensure a more global perspective.
Expand Author Diversity:
Reference a broader array of researchers to include diverse viewpoints and avoid potential biases.
Standardize Reference Formatting:
Ensure all references are formatted consistently according to a specific citation style (e.g., APA, MLA).
Focus on Directly Relevant Studies:
Prioritize references that directly address the relationship between virtual simulation, critical thinking, and self-directed learning in nursing education.
Peer-reviewed Sources:
Emphasize peer-reviewed journal articles and exclude less reliable sources, such as non-peer-reviewed dissertations.
Strengthen Evidence Base:
Seek out and include additional studies that robustly support the claims made in the study about the benefits of virtual simulation.
By addressing these drawbacks, the study can be supported by a more robust, reliable, and current set of references, enhancing the overall credibility and impact of the research.

Experimental design

Methodology feedback:
Aim of the study is not clear enough
• Study questions are missed.
Sample Size Determination:
Unclear Rationale: The methodology mentions that the sample size of 201 was calculated based on the Epi Info program, with a confidence level of 99% and a sampling error of 1%. However, it does not provide the specific population size or how these parameters were chosen, which makes it difficult to assess whether the sample size is adequate.
Data Collection Technique:
Reliance on Self-Reported Data: The study relies on self-reported data collected through a Google Forms questionnaire. Self-reported data can be subject to various biases, such as social desirability bias, recall bias, and response bias, which can affect the accuracy and reliability of the data.
Single Institution Focus:
Limited Context: The study is conducted at a single institution (Princess Nourah bint Abdulrahman University). This focus on one university limits the variability in the educational environment and curriculum, which could influence the results. This again affects the generalizability of the findings.
Instruments of Data Collection:
Lack of Detail: The methodology mentions adopting three scales for data collection but does not provide specific details about these scales, such as their validity, reliability, or how they were adapted. Detailed information on the instruments is necessary to evaluate the appropriateness and accuracy of the measurement tools.
Exclusion Criteria:
Potential Overlooked Factors: Excluding first and second-year nursing students who did not experience simulation could be a drawback if these students could provide valuable comparative data on the impact of simulation experiences throughout different stages of their education.
Clarification of Study Aim:
Potential Influence on Responses: The methodology mentions that the aim and content of the study were clarified to the participants before data collection. While this is important for informed consent, it might influence the participants' responses, knowing they are being studied for their critical thinking and SDL abilities related to virtual simulation.
Suggestions for Improvement
Sampling Technique:
Consider using a probability sampling technique to improve the representativeness of the sample and reduce selection bias.
Sample Size Justification:
Provide a clear explanation of the population size, the rationale for choosing the confidence level and sampling error, and how these parameters were used to calculate the sample size.
Data Collection Methods:
Combine self-reported data with objective measures of critical thinking and SDL abilities, such as performance assessments or observational data, to enhance the validity of the findings.
Multiple Institutions:
Conduct the study across multiple institutions to increase the generalizability of the results and account for variations in educational environments and curricula.
Instrument Details:
Provide detailed information on the scales used for data collection, including their development, adaptation process, validity, and reliability.
Broader Inclusion Criteria:
Consider including first and second-year students to provide a more comprehensive view of the impact of virtual simulation across different stages of nursing education.
Minimize Influence on Responses:
While ensuring informed consent, minimize detailed disclosure of study aims that might bias participants' responses, and consider employing techniques to reduce social desirability and other biases.

Validity of the findings

Conclusion:
Despite The conclusion of the study effectively summarizes the main findings and their implications for nursing education. However, it does have several drawbacks:
Lack of Specificity:
The conclusion makes broad statements about the benefits of virtual simulation but lacks specific details or examples from the study findings. Including specific statistics or notable results would strengthen the conclusion.
Redundancy:
Some points are repeated, such as the benefit of virtual simulation in developing critical thinking and self-learning abilities. This redundancy could be reduced to make the conclusion more concise.
Overgeneralization:
The conclusion implies that virtual simulation will be universally beneficial for all nursing students and educators. This overgeneralizes the findings, especially given the study’s limitations (e.g., limited sample size, single-gender participants, single institution).
Lack of Acknowledgment of Limitations:
The conclusion does not mention the limitations of the study, such as the restricted sample size, use of non-probability convenience sampling, and focus on a single university. Acknowledging these limitations is crucial for a balanced and transparent conclusion.
No Future Directions:
The conclusion does not suggest directions for future research, which is essential for building on the current study’s findings. It would be beneficial to recommend further studies with larger, more diverse samples or different methodologies.
Educational Recommendations:
While it briefly mentions the potential benefits for nursing educators, the conclusion does not provide concrete recommendations or strategies for implementing virtual simulations in nursing curricula.
Summary of Drawbacks
Sample Limitation: Only female participants from one university.
Sampling Technique: Non-probability convenience sampling limits generalizability.
Lack of Diversity: Results cannot be generalized to a wider population without further research.
Conclusions and Implications:
Causal Inference: The abstract suggests that virtual simulation "made" students practice critical thinking and self-learning, implying causation, which cannot be concluded from a correlational study.

Additional comments

Abstract
This abstract presents valuable findings on the relationship between virtual simulation and critical thinking as well as self-directed learning abilities among nursing students. However, several drawbacks can be identified:
Questionnaire Adaptation: There is no information on how the questionnaire was adapted, whether it was validated for the specific population, or what specific questions were included.
Insufficient Description of Participants:
Demographics: The abstract does not provide any demographic information about the participants (e.g., age, gender, prior experience with virtual simulations), which could influence the results and their interpretation.
Year of Study: While it mentions third- and fourth-year students, it does not differentiate between the results for these groups, which might have different levels of exposure and experience.
Statistical Analysis Details:
Missing Statistical Values: Although the abstract provides p-values and correlation coefficients, it lacks other important statistical details such as confidence intervals, effect sizes, or how assumptions of the statistical tests were checked.
Interpretation of p-values: The abstract states significant relationships with specific p-values but does not discuss the clinical or practical significance of these findings.
Results Presentation:
Percentages and Strength of Agreement: The percentages of agreement (56% agreed, 27% strongly agreed) are provided without contextualizing what constitutes "agree" versus "strongly agree" and how these were quantified.
Positive Relationships: The terms "strong positive relationship" and "significant positive relationship" are used without a clear definition of what constitutes "strong" in the context of correlation coefficients (e.g., a coefficient of 0.65 versus 0.78).

Future Research: There is no discussion on the limitations of the study or suggestions for future research to build on these findings or address the identified limitations.
where is the significance of the study?
what about the theoretical framework that guide your study?
why 3rd and 4th year only?
is it enough to test for critical disposition? i think definition of study variables will add benefit to the study and it will be clear for readers.
one thing more how did authors you guarantee the informed consent from students??
Addressing these drawbacks would strengthen the study’s credibility and provide a clearer, more comprehensive understanding of the relationship between virtual simulation, critical thinking, and self-directed learning abilities among nursing students.

Reviewer 2 ·

Basic reporting

General comments
The title of the article The relationship between virtual simulation, critical thinking and self-directed learning ability of nursing students in Riyadh, Saudi Arabia is very interesting and related to an interesting topic. The title correctly explains the purpose and objective of the article.
The summary of the article is structured and contains the necessary parts: objective, methods, results and conclusion. The abstract contains an accurate summary of the article, the language used in the abstract is easy to read. The authors provide appropriate background on the topic and rationale of this article and describe what the authors hope to achieve. Keywords (sort alphabetically).

In the introduction, the authors clearly and concisely describe the relationship between virtual simulation, critical thinking and self-directed learning abilities of nursing students. I suggest that the authors provide more previous empirical studies that support the idea behind the implementation or demonstrate the significance of the current study.The research design is described in detail.

The research design is adequate and there are no particular shortcomings.
The measuring instrument is clearly described.
The population of interest and the sampling procedure are clearly defined.
The data collection procedure is clearly described.

Results: the results are clearly presented, the authors provide accurate research results, and there is sufficient evidence for each result.
Discusion: Give more studies to discuss your findings.

The conclusion is clear.
Finally, this was interesting information about the article. In its current state, it adds many new insights.
Basic reporting
The article is written in professional English (AME Certificate-Simulation Manuscript)
Literature, enough background/context of the field.
The article contains enough introduction and background to show how the paper fits into the wider body of knowledge. Relevant previous literature is cited appropriately.

Experimental design

Research question well defined, relevant and meaningful. It was stated that the research fills a perceived gap in knowledge about teaching nursing students.
The research question is a clear question. The authors identified a relationship between virtual simulation, critability to think and self-directed learning
nursing students in Riyadh, Saudi Arabia.
A rigorous investigation conducted in accordance with high technical and ethical standards.
The research was conducted in accordance with the applicable ethical standards in the field.
The authors stated the limitations of the study (appropriate number of subjects and sampling method), inclusion factors and exclusion factors.

Methods described with sufficient detail and information for replication.
The methods are described with enough information that another examiner can repeat them.

Experimental design

Experimental design

Research question well defined, relevant and meaningful. It was stated that the research fills a perceived gap in knowledge about teaching nursing students.
The research question is a clear question. The authors identified a relationship between virtual simulation, critnn ability to think and self-directed learning nursing students in Riyadh, Saudi Arabia.

Validity of the findings

A rigorous investigation conducted in accordance with high technical and ethical standards.
The research was conducted in accordance with the applicable ethical standards in the field.
The authors stated the limitations of the study (appropriate number of subjects and sampling method), inclusion factors and exclusion factors.

Reviewer 3 ·

Basic reporting

the English language is clear and the literature references sufficient

Experimental design

this part not applicable since the research design is descriptive

Validity of the findings

no comments

Additional comments

Thank you for the opportunity to review your research article. Your study addresses an important area of nursing education, and I appreciate the work you have done. I would like to offer the following comments to help improve the clarity and quality of your manuscript:
1. Reference Style (Page 7, Line 56): The reference style for “Foronda et al. (2020)” should be consistent with the rest of the manuscript. Please use the reference number [11] to maintain uniformity throughout the document.
2. Abbreviation (Page 8, Line 74): Please provide the full form of "PNU" when first mentioned and then use the abbreviation consistently throughout the text.
3. Material and Methods - Study Design (Page 8, Line 78): The aim of the study should be removed from the "Study Design" section. Instead, it would be more appropriate to place it in the introduction or after study design with title aim of the study. Additionally, please confirm whether the study's aim is to "investigate the correlation between the variables of virtual simulation, self-directed learning abilities, and critical thinking skills among nursing students," as suggested, or if it is as stated in the abstract: "to assess the relationship between virtual simulation and critical thinking disposition and self-directed learning abilities among nursing students" (Page 5, Line 7). Consistency in the aim is crucial.
4. Study Population: It would be helpful to include details on the total number of students in the BSN Year 3 and Year 4. Please clarify whether all eligible students participated in the study or if any declined.
5. Simulation Details: The manuscript would benefit from additional information about the type of simulation used. Was it high-fidelity or purely virtual? A more detailed description of the virtual simulation, including the techniques used and how the sessions were conducted, would provide valuable context for the readers.
6. References: Please consider updating the references, as several cited works are quite old (e.g., from 2001, 2004). Including more recent literature would strengthen the foundation of your study.

Thank you

---

## Round 0.2 · accepted · Accept

Thank you for making edits based on reviewer feedback.